# Generation of a Triple-Transgenic Zebrafish Line for Assessment of Developmental Neurotoxicity during Neuronal Differentiation

**DOI:** 10.3390/ph12040145

**Published:** 2019-09-24

**Authors:** Junko Koiwa, Takashi Shiromizu, Yuka Adachi, Makoto Ikejiri, Kaname Nakatani, Toshio Tanaka, Yuhei Nishimura

**Affiliations:** 1Department of Integrative Pharmacology, Mie University Graduate School of Medicine, Tsu, Mie 514-8507, Japan; j-koiwa@doc.medic.mie-u.ac.jp (J.K.); tshiromizu@doc.medic.mie-u.ac.jp (T.S.); 319m001@m.mie-u.ac.jp (Y.A.); 2Department of Central Laboratory, Mie University Hospital, Tsu, Mie 514-8507, Japan; mktik@clin.medic.mie-u.ac.jp; 3Department of Genomic Medicine, Mie University Hospital, Tsu, Mie 514-8507, Japan; nakatani@clin.medic.mie-u.ac.jp; 4Department of Systems Pharmacology, Mie University Graduate School of Medicine, Tsu, Mie 514-8507, Japan; tanaka@doc.medic.mie-u.ac.jp

**Keywords:** developmental neurotoxicity, neuronal differentiation, zebrafish, in vivo fluorescence imaging

## Abstract

The developing brain is extremely sensitive to many chemicals. Exposure to neurotoxicants during development has been implicated in various neuropsychiatric and neurological disorders, including autism spectrum disorders and schizophrenia. Various screening methods have been used to assess the developmental neurotoxicity (DNT) of chemicals, with most assays focusing on cell viability, apoptosis, proliferation, migration, neuronal differentiation, and neuronal network formation. However, assessment of toxicity during progenitor cell differentiation into neurons, astrocytes, and oligodendrocytes often requires immunohistochemistry, which is a reliable but labor-intensive and time-consuming assay. Here, we report the development of a triple-transgenic zebrafish line that expresses distinct fluorescent proteins in neurons (Cerulean), astrocytes (mCherry), and oligodendrocytes (mCitrine), which can be used to detect DNT during neuronal differentiation. Using in vivo fluorescence microscopy, we could detect DNT by 6 of the 10 neurotoxicants tested after exposure to zebrafish from 12 h to 5 days’ post-fertilization. Moreover, the chemicals could be clustered into three main DNT groups based on the fluorescence pattern: (i) inhibition of neuron and oligodendrocyte differentiation and stimulation of astrocyte differentiation; (ii) inhibition of neuron and oligodendrocyte differentiation; and (iii) inhibition of neuron and astrocyte differentiation, which suggests that reporter expression reflects the toxicodynamics of the chemicals. Thus, the triple-transgenic zebrafish line developed here may be a useful tool to assess DNT during neuronal differentiation.

## 1. Introduction

The developing brain is more vulnerable than the adult brain to most chemicals [1,2,3,4,5,6]. Exposure to neurotoxicants during development has been implicated in various neurodevelopmental and neuropsychiatric diseases, including autism spectrum disorders, attention deficit hyperactive disorder, learning disabilities, and schizophrenia [7,8,9,10,11,12,13,14,15]. The Organisation for Economic Co-operation and Development (OECD) has developed guidelines to assess the developmental neurotoxicity (DNT) of chemicals, with rodents as the model animal of choice [16]. However, while rodents have contributed significantly to our understanding of DNT [3,17,18,19], experiments using large numbers of rodents are time-consuming, expensive, and accompanied by ethical concerns. Thus, alternative test systems, such as human stem/progenitor cell and zebrafish models, have the potential to enable testing of large chemical libraries while reducing expense and minimizing rodent use [4,20,21,22,23,24,25].

In human stem/progenitor cell models, various endpoints have been used to assess DNT, including viability, apoptosis, proliferation, migration, differentiation, and neuronal network formation [24,26,27]. These endpoints facilitate the identification of adverse outcome pathways and the development of integrated approaches to test and assess chemical-induced DNT [22,28]. Differentiation of neural stem/progenitor cells into neurons, astrocytes, and oligodendrocytes has been analyzed by immunohistochemical staining with antibodies against cell-specific proteins such as NeuN/RNA-binding Fox3 (Rbfox3) for neurons, glial fibrillary acidic protein (GFAP) for astrocytes, and myelin basic protein (MBP) for oligodendrocytes [26,29]. Reporter assays in which expression of fluorescent proteins is driven by the promoters of genes selectively expressed in neurons, astrocytes, and oligodendrocytes have also been used to assess the effects of developmental neurotoxicants on cell differentiation [30]. Fluorescent reporter assays do not require time-consuming procedures such as fixation and immunostaining, and they can be performed in a relatively high-throughput manner. However, distinguishing between multiple cell types requires a multiplexed fluorescent reporter system and, although this has been employed in some human stem/progenitor cell models [31], to our knowledge, it has not been used to distinguish between neurons, astrocytes, and oligodendrocytes in the same organism.

Compared with other species, zebrafish offer several advantages as a model system for DNT testing. For example, the pattern of developmental gene expression and structure of various brain regions is relatively conserved in zebrafish; a wide range of chemicals can be absorbed from the surrounding medium; the small size and prolific breeding capacity lowers housing and experimental costs [4,23,32,33,34,35]; and the optical transparency, particularly of pigmentless mutants, make zebrafish suitable for in vivo fluorescence imaging [4,20]. To date, many transgenic (Tg) zebrafish lines have been developed that express fluorescent proteins in specific neuronal subtypes [36,37,38], and multiplexed reporter systems have also been implemented in this species [39]. To facilitate DNT testing, we have developed a triple-fluorescent Tg zebrafish line in which progenitor cell differentiation into neurons, astrocytes, and oligodendrocytes is accompanied by cell type-specific expression of the fluorescent proteins Cerulean, mCherry, and mCitrine, respectively, thereby enabling the effects of neurotoxicants on neuronal differentiation to be assessed in vivo.

## 2. Results

### 2.1. Construction of Transposon Vectors to Express Cerulean, Mcherry, and Mcitrine in Neurons, Astrocytes, and Oligodendrocytes

We selected the Tol2 system to selectively express the three fluorescent proteins in neurons, astrocytes, and oligodendrocytes [40]. Tol2 is an autonomous transposon that can catalyze transposition of a DNA construct flanked by Tol2 sequences into the host genome. Cerulean, mCherry, and mCitrine were selected as the reporter proteins because their fluorescent signals can be distinguished using CFP, RFP, and YFP filters. To direct cell type-specific reporter expression, we used the regulatory sequences of enolase 2 (eno2), gfap, and mbp, which are selectively expressed in neurons, astrocytes, and oligodendrocytes, respectively [36,37,38]. The regulatory sequence of eno2 gfap, or mbp was cloned upstream of the coding region of Cerulean, mCherry, or mCitrine, respectively, and then inserted between two Tol2 sequences in the vector backbone, generating eno2: Cerulean, gfap:mCherry, and mbp:mCitrine vectors (Figure 1).

The coding regions of Cerulean, mCherry, and mCitrine were placed downstream of the regulatory sequences of eno2, gfap, and mbp, and cloned between two Tol2 sequences in the vector backbone to direct selective protein expression in neurons, astrocytes, and oligodendrocytes, respectively. The resulting Tg (eno2:Cerulean, gfap:mCherry, mbp:mCitrine) zebrafish line is referred to as the triple-Tg line.

### 2.2. Generation of Triple-Tg Zebrafish

One of the three transposon vectors was injected together with transposase mRNA into fertilized zebrafish eggs to generate three lines expressing a single fluorescent protein (Cerulean, mCherry, or mCitrine) in the central nervous system (CNS). At maturity, the single-Tg zebrafish were mated to generate double-Tg zebrafish. In turn, the adult double-Tg zebrafish were mated to generate triple-Tg zebrafish expressing Cerulean, mCherry, and mCitrine in the CNS. Representative in vivo images of the larvae at 5 days’ post-fertilization (dpf) is shown in Figure 2. The fluorescence patterns obtained are consistent with the results of previous studies of single fluorescent protein expression in zebrafish neurons, astrocytes, or oligodendrocytes driven by eno2, gfap, or mbp promoters, respectively [36,37,38]. These results suggest that the fluorescent proteins are selectively expressed in neurons, astrocytes, and oligodendrocytes.

Representative in vivo fluorescence images of triple-Tg zebrafish at 5 dpf.

### 2.3. Assessment of DNT Using Triple-Tg Zebrafish

To assess the DNT of a panel of chemicals, we first measured the no observed effect concentration (NOEC) for lethality, which is the highest concentration of each chemical to which zebrafish are exposed that does not significantly affect the lethality compared with the controls. Triple-Tg zebrafish were exposed to log serial dilutions of 13 chemicals from 12 h post-fertilization (hpf) to 5 dpf and then imaged at 5 dpf. The effects of the chemicals on lethality were evaluated according to a Fish Embryo Acute Toxicity Test guideline (OECD TG236) [41]. The measured NOEC for lethality of each chemical was designated the maximum tolerable concentration (MTC) and was used for DNT assessment. We employed 13 chemicals considered suitable for DNT test method validation [42,43], of which 10 were established developmental neurotoxicants: valproic acid (VPA), trichostatin A (TSA), carbamazepine (CBZ), nicotine (NCT), chlorpyrifos (CPF), cyclopamine (CPM), methyl mercury (MeHg), dexamethasone (DEX), retinoic acid (RA), and bisphenol A (BPA); and 3 were not neurotoxicants: deferoxamine (DFX), saccharin (SAC), and acetaminophen (APAP). We selected the 10 developmental neurotoxicants because they have been reported to affect neuronal differentiation [44,45,46,47,48,49,50,51,52,53,54,55,56]. The MTCs for these chemicals were 1 nM for RA, 100 nM for TSA, CPM, and MeHg, 10 μM for CPF, 100 μM for VPA, CBZ, DEX, and SAC, and 1000 μM for DFX and APAP (Figure 3).

We next performed in vivo fluorescence imaging of the triple-Tg zebrafish after exposure to the 13 chemicals at their MTCs from 12 hpf to 5 dpf. The fluorescent signals were then quantified and normalized to the values for untreated control zebrafish (Figure 4 and Appendix A). We found that Cerulean fluorescence (CFP) was significantly reduced by exposure of zebrafish to TSA, CBZ, NCT, BPA, and CPF (Figure 4A and Appendix A), whereas mCherry fluorescence (RFP) was significantly increased by VPA and TSA and significantly decreased by BPA (Figure 4B and Appendix A), and mCitrine fluorescence (YFP) was significantly decreased by VPA, TSA, CBZ, and NCT (Figure 4C and Appendix A). To translate the fluorescent protein expression patterns into functional effects on neuronal differentiation, we calculated the ratios of the fluorescent signals. VPA, TSA, and CBZ significantly increased the RFP/CFP ratio and decreased the YFP/CFP and YFP/RFP ratios (Figure 4D–F and Appendix A), suggesting that these compounds increased and decreased the differentiation of progenitors into astrocytes and oligodendrocytes, respectively. NCT significantly decreased the ratios of YFP/CFP and YFP/RFP, but not RFP/CFP (Figure 4D–F and Appendix A), suggesting that NCT preferentially interfered with the differentiation into oligodendrocytes. SAC, APAP, and DFX did not significantly affect the fluorescent signals, which is consistent with the fact that these chemicals are considered to not be neurotoxicants. However, we were not able to detect the significant change of these fluorescent signals in zebrafish exposed to CPM, RA, MeHg, and DEX that have been reported to be developmental neurotoxicants [42,43]. Hierarchical clustering of the fluorescence parameters for each chemical (Figure 5) revealed tight clustering of (i) TSA, VPA, and CBZ; (ii) CPM, NCT, and CPF; (iii) BPA and RA; and (iv) DEX, SAC, APAP, and DFX; reflecting on the trend of (i) increase of RFP and decrease of YFP and CFP; (ii) decrease of YFP and CFP; (iii) decrease of RFP and CFP; and (iv) little change of CFP, RFP, and YFP, respectively. MeHg, whose average signal of YFP was relatively high, was not tightly clustered with other chemicals. Thus, the triple-Tg zebrafish line appears to be capable of discriminating between developmental neurotoxicants that affect different cell types within the CNS.

## 3. Discussion

Here, we report the development of a triple-Tg zebrafish line expressing Cerulean, mCherry, and mCitrine in neurons, astrocytes, and oligodendrocytes, respectively, and we demonstrate with in vivo fluorescence imaging that the line can be used to assess the effects of chemicals on the differentiation of neural progenitors into the three cell types. Thus, this triple-Tg zebrafish line may be a useful tool for DNT testing during neuronal differentiation.

Previous work has demonstrated the utility of Tg zebrafish lines expressing a single fluorescent protein in neurons for DNT testing. For example, animals expressing green fluorescent protein (GFP) under the control of the neuron-specific hb9 promoter were successfully used to demonstrate a dose-dependent reduction of axon length following exposure to ethanol [57], which is consistent with the DNT of ethanol in humans [58,59]. Similarly, a zebrafish line with nkx2.2a-driven expression of GFP in motor neurons was used to demonstrate significant shortening of axons following exposure to five neurotoxicants at concentrations that did not cause external malformations [60]. Another Tg zebrafish line expressing mCitrine in oligodendrocytes under the control of the mbp promoter was able to demonstrate not only the DNT of antithyroid drugs (methimazole and propylthiouracil), but also the stimulatory effects of thyroid hormone on oligodendrocyte differentiation [61]. A zebrafish line with HuC/elavl3-driven expression of GFP in neurons and olig2-driven dsRed expression in oligodendrocytes was successfully used to detect DNT of trimethyltin chloride [62]. To our knowledge, however, there have been no reports of DNT testing using a triple-Tg zebrafish line expressing distinct fluorescent proteins in neurons, astrocytes, and oligodendrocytes.

Here, we selected 13 chemicals recommended as reference compounds to validate newly developed assays for DNT [42,43]; of which 10 have been reported to be neurotoxicants and 3 have no reported neurotoxicity. By quantifying the relative fluorescence of each reporter protein in chemical-exposed zebrafish, we were able to detect significant effects of 6 of the 10 putative neurotoxicants (VPA, TSA, CBZ, NCT, CPF, and BPA) on neuronal differentiation.

Based on their fluorescent signatures, VPA, TSA, and CBZ appeared to inhibit and stimulate the differentiation of oligodendrocytes and astrocytes, respectively. These results are consistent with previous studies in rodents demonstrating that VPA inhibits and stimulates neural progenitor cell differentiation to oligodendrocytes and astrocytes, respectively [44,45]. Activation of histone deacetylase 3 (HDAC3) has been reported to stimulate and inhibit the differentiation of rat neural progenitors to oligodendrocytes and astrocytes, respectively [63]. Interestingly, VPA, TSA, and CBZ are known to inhibit HDAC3 [64,65,66], suggesting that they may affect oligodendrocyte and astrocyte differentiation by suppressing HDAC3 activity in neural progenitor cells. Our results are also consistent with the report demonstrating the inhibition of neurogenesis by TSA and other HDAC inhibitors [67].

We found that exposure of the triple-Tg zebrafish to NCT, a nicotinic acetylcholine receptor (nAChR) agonist, suppressed the differentiation of oligodendrocytes, as indicated by the fluorescent signatures. This finding is consistent with previous work showing that gestational exposure of rats to NCT resulted in significantly reduced expression of myelin genes, including Mbp, in the prefrontal cortex and nucleus accumbens of juvenile rats [46]. We also found that CPF, an acetylcholinesterase (AChE) inhibitor, showed a trend to suppress the differentiation of oligodendrocytes in the triple-Tg zebrafish. In a similar study, exposure of rats to the AChE inhibitor carbofuran from gestational day 7 onwards also led to impaired myelination in the hippocampus on postnatal day 21 [47]. The results of the present study are consistent with these reports, collectively suggesting that misregulation of nAChR function during development may be associated with impaired oligodendrocyte differentiation. Our results are also consistent with the report demonstrating the inhibition of neurogenesis by NCT and CPF [68,69]. Hierarchical clustering of the fluorescence parameters revealed that CPM was tightly clustering with NCT and CPF. It is noteworthy that CPM inhibits PAX6 and hedgehog signaling, key players in the differentiation into neurons and oligodendrocytes, respectively [48,70,71].

Triple-Tg zebrafish exposed to BPA showed significantly decreased Cerulean and mCherry expression, indicative of reduced the differentiation into neurons and astrocytes, respectively. BPA attenuate the expression of Pax6 [49], suggesting that BPA may decrease the differentiation into neurons by disrupting Pax6 functions. The mechanisms of how BPA decrease the differentiation into astrocytes remain to be elucidated. RFP signal seems to be reduced by RA in the hierarchical clustering. This is consistent with a previous report demonstrating that RA suppressed the expression of GFAP in embryonic day 13 cortical progenitor cells [55], although the reduction was not statistically significant in this study.

We were not able to detect significant effects of MeHg and DEX on the neuronal differentiation in the triple-Tg zebrafish. It has been demonstrated that MeHg, DEX, and RA significantly affect neural progenitor differentiation in rodents [55,56,72]. In this study, zebrafish were exposed to these chemicals from 12 hpf to 5 dpf. We can change the exposure time and concentration of chemicals. Further studies are needed to optimize the assessment of DNT using triple-Tg zebrafish. The optimization could also stimulate using the triple-Tg zebrafish for drug discovery targeting neural regeneration and other studies focusing on neuronal differentiation.

Finally, developmental exposure of triple-Tg zebrafish to SAC, APAP, and DFX had no significant effects on fluorescent protein expression, which is consistent with previous demonstrations that these chemicals are not developmental neurotoxicants [42,43].

This study has several limitations. First, we used eno2, gfap, and mbp as markers for neurons, astrocytes, and oligodendrocytes, respectively. Although these genes are well-established neuronal cell type-specific markers in mammals [73,74,75] and zebrafish [36,37,38], they are expressed at relatively late stages of neuronal differentiation [73,74,75]. Therefore, the triple-Tg zebrafish developed in this study may not be suitable for DNT testing in the early stages of neuronal differentiation. Second, zebrafish can absorb most, but not all, chemicals from the surrounding medium, and chemicals with low lipophilicity (high hydrophilicity) tend to be poorly absorbed [4,33,76]. Third, the molecular mechanisms underlying the DNT of each chemical on neuronal differentiation in the Tg zebrafish remain to be clarified. Genome editing technology is easily performed in zebrafish, making it feasible to generate gene-specific knockout animals. Integrative analysis of various knockout and transgenic zebrafish lines will make it possible to clarify the adverse outcome pathways underlying the effects of neurotoxicants on neuronal differentiation.

In summary, we have generated a triple-Tg zebrafish line to enable DNT screening of neuron, astrocyte, and oligodendrocyte differentiation within 1 week. Further studies with an expanded chemical library will be required to rigorously assess the utility of in vivo fluorescent imaging of triple-Tg zebrafish as an alternative method to assess DNT.

## 4. Materials and Methods

### 4.1. Ethics Statement

Mie University Institutional Animal Care and Use Committee guidelines state that no approval is required for experiments using zebrafish. Nonetheless, the animal experiments described in this manuscript conform to the ethical guidelines established by the committee.

### 4.2. Compounds

Valproic acid (VPA), deferoxamine (DFX), acetaminophen (APAP), nicotine (NCT), dexamethasone (DEX), chlorpyrifos (CPF), and methyl mercury (MeHg) were purchased from Sigma (St. Louis, MO, USA). Cyclopamine (CPM) and retinoic acid (RA) were purchased from Wako (Osaka, Japan). Trichostatin A (TSA), carbamazepine (CBZ), bisphenol A (BPA), and saccharin (SAC) were purchased from Tokyo Kasei (Tokyo, Japan). With the exception of CPM, DFX, and APAP, stock solutions were prepared in dimethyl sulfoxide (DMSO; Nacalai, Kyoto, Japan). CPM was dissolved in ethanol (Wako), and DFX and APA were dissolved in in 0.3× Danieau’s solution (19.3 mM NaCl, 0.23 mM KCl, 0.13 mM MgSO_4_, 0.2 mM Ca(NO_3_)_2_, 1.7 mM HEPES, pH 7.2). For experiments, chemicals were serially diluted in 0.3× Danieau’s solution. Controls contained the same final concentrations of vehicle.

### 4.3. Zebrafish Husbandry

Zebrafish were maintained according to standard methods as described previously [61,77]. Briefly, zebrafish were raised at 28.5 ± 0.5 °C with a 14/10 h light/dark cycle. Embryos were obtained by natural mating and cultured in 0.3× Danieau’s solution.

### 4.4. Generation of Tg (eno2:Cerulean, gfap:mCherry, mbp:mCitrine) Zebrafish

We used an albino zebrafish line [78] (Max Planck Institute for Developmental Biology, Tübingen, Germany) to generate the Tg zebrafish line. The coding regions of Cerulean, mCherry, and mCitrine were amplified by PCR from pCS2+8NCerulean, pCS2+8NmCherry, and pCS2+8NmCitrine plasmids, respectively (Addgene, Cambridge, MA, USA), and cloned into a Tol2 vector using the In-fusion HD cloning kit (Takara Bio, Shiga, Japan) to generate three circular plasmids (pT2-Cerulean, pT2-mCherry, and pT2-mCitrine). Briefly, a fragment (bp 3017 to 1088) of pT2AL200R150G [40] was amplified by inverse PCR and fused with the coding regions of Cerulean, mCherry, or mCitrine to generate the circular plasmids. The promoters of zebrafish eno2 (−3783 to −3723 bp) [36] and mbp (−1873 to 80 bp) [38] were synthesized (Invitrogen, Carlsbad, CA, USA) and cloned into the relevant pT2 plasmid using the In-fusion HD cloning kit (Takara Bio, Shiga, Japan). The promoter of zebrafish gfap [37] was amplified from pEGFP-gfap (Intron1/5′/Exon1-zebrafish; Addgene) by PCR and cloned into pT2-mCherry using the In-fusion HD cloning kit (Takara Bio). To generate single-Tg zebrafish, one of the three plasmids and transposase mRNA were injected into zebrafish embryos at the 1–4 cell stage, and larvae expressing the fluorescent protein in the CNS were selected and maintained. Mature F0 single-Tg zebrafish were mated with albino zebrafish and single-Tg F1 animals were selected and maintained. Mature F1 male single-Tg zebrafish were mated with mature F1 female single-Tg zebrafish expressing a different fluorescent protein to generate F2 double-Tg zebrafish. Mature F2 double-Tg male and female zebrafish (e.g., males expressing Cerulean and mCherry and females expressing mCherry and mCitrine) were mated to generate F3 zebrafish. Finally, the F3 zebrafish with eno2-driven Cerulean expression in neurons, gfap-driven mCherry expression in astrocytes, and mbp-driven mCitrine expression in oligodendrocytes were selected by in vivo fluorescence imaging at 5 dpf, maintained, and analyzed.

### 4.5. Exposure of Triple-Tg Zebrafish to Chemicals

Mature triple-Tg zebrafish were mated with the albino mutant line and the resulting embryos (40 per well in 6-well plates) were exposed to serial dilutions of the chemicals from 12 hpf to 5 dpf without changing the medium. zebrafish were then imaged using a nikon smz800 stereoscopic microscope according to a fish embryo acute toxicity test guideline (oecd tg236) [41]. the highest chemical concentration that did not induce lethality (noec) was taken as the maximum tolerable concentration (mtc) for assessment of dnt.

### 4.6. In Vivo Imaging of Triple-Tg Zebrafish

Mature triple-Tg zebrafish were mated with the albino mutant line, and the embryos (40/well in 6-well plates) were exposed to chemicals at the MTCs from 12 hpf to 5 dpf without changing the medium. At 5 dpf, the larvae were transferred to fresh 0.3× Danieau’s solution containing 2-phenoxyethanol (500 ppm) to be anesthetized and then transferred to glass slides. A few drops of 3% low-melting agarose solution were placed over the larvae, and the animals were immediately oriented with the dorsal side up. The zebrafish were then observed using an epifluorescence microscope (SMZ25, Nikon, Tokyo, Japan) with the following filters: CFP (Ex/Em 425–445/457–500 nm), RFP (Ex/Em 530–560/590–650 nm), and YFP (Ex/Em 458–512/529–550 nm) to detect Cerulean (Ex/Em 433/475 nm), mCherry (Ex/Em 587/610 nm), and mCitrine (Ex/Em 516/529 nm) fluorescence, respectively. Fluorescent signals were quantified using Volocity (Perkin Elmer, Cambridge, MA, USA). A region of interest was placed around the Cerulean fluorescence observed in the brain and spinal cord in the 256 gray-scale (0–255) image. The areas of CFP, RFP, and YFP fluorescence within the region of interest that contained pixels above the intensity threshold (30 for Cerulean and mCherry, 20 for mCitrine) were measured. Single-protein signals and the ratios of RFP/CFP, YFP/CFP, and YFP/RFP signals were calculated.

### 4.7. Statistical Analysis

CFP, RFP, YFP, RFP/CFP, YFP/CFP, and YFP/RFP fluorescent signals in each treated zebrafish were normalized to the average signals in the untreated control group. Data are shown as the mean ± standard error of the mean (SEM). We performed D’Agostino–Pearson normality test to examine the distribution of these data related to each chemical. The test revealed that these data were not always normally distributed. We, therefore, used Kruskal–Wallis test and Dunnett’s multiple comparisons test to examine the differences between group means. Hierarchical clustering of log-transformed normalized fluorescent signals was performed using Heatmapper [79] with Manhattan as the distance measurement method and average linkage as the clustering method. Statistical analyses were performed using Prism 7 software (GraphPad, La Jolla, CA, USA).

## Figures and Tables

**Figure 1 pharmaceuticals-12-00145-f001:**
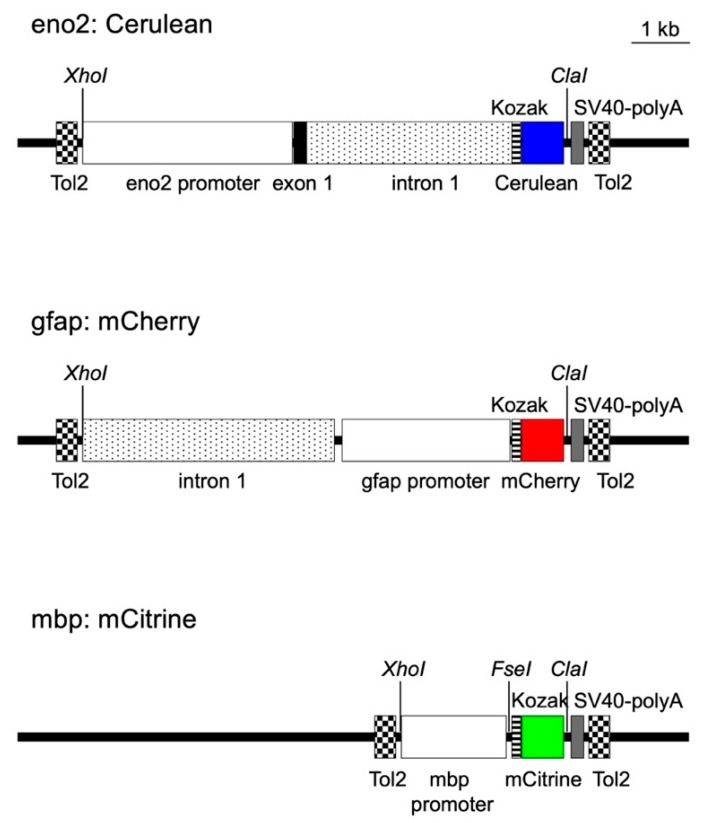
Schematic representation of transposon vectors.

**Figure 2 pharmaceuticals-12-00145-f002:**
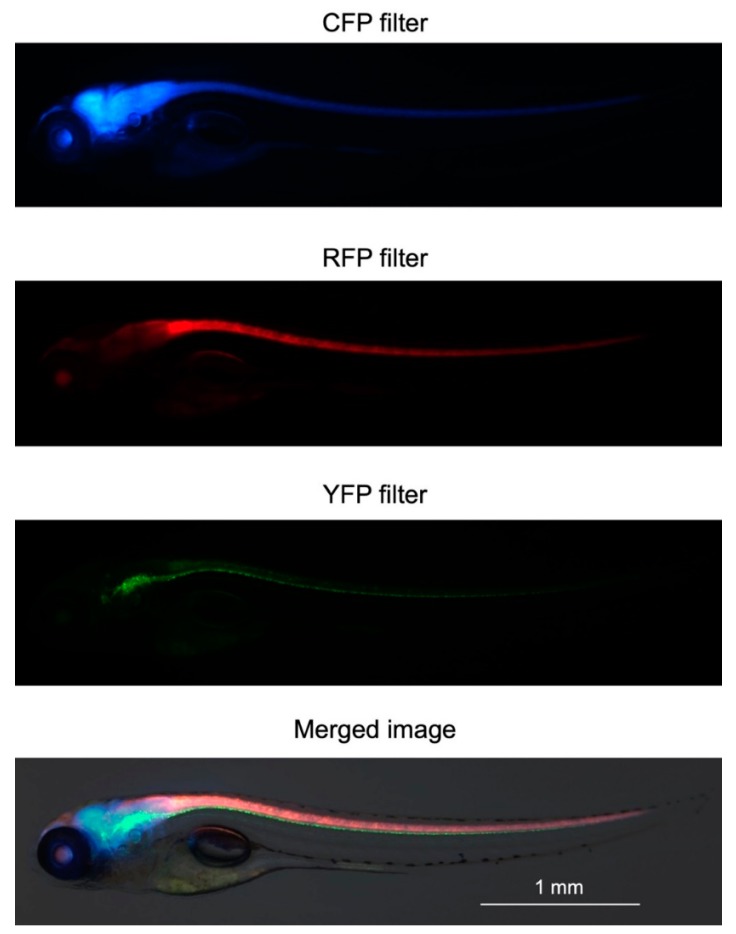
In vivo fluorescence imaging of the triple-Tg zebrafish line.

**Figure 3 pharmaceuticals-12-00145-f003:**
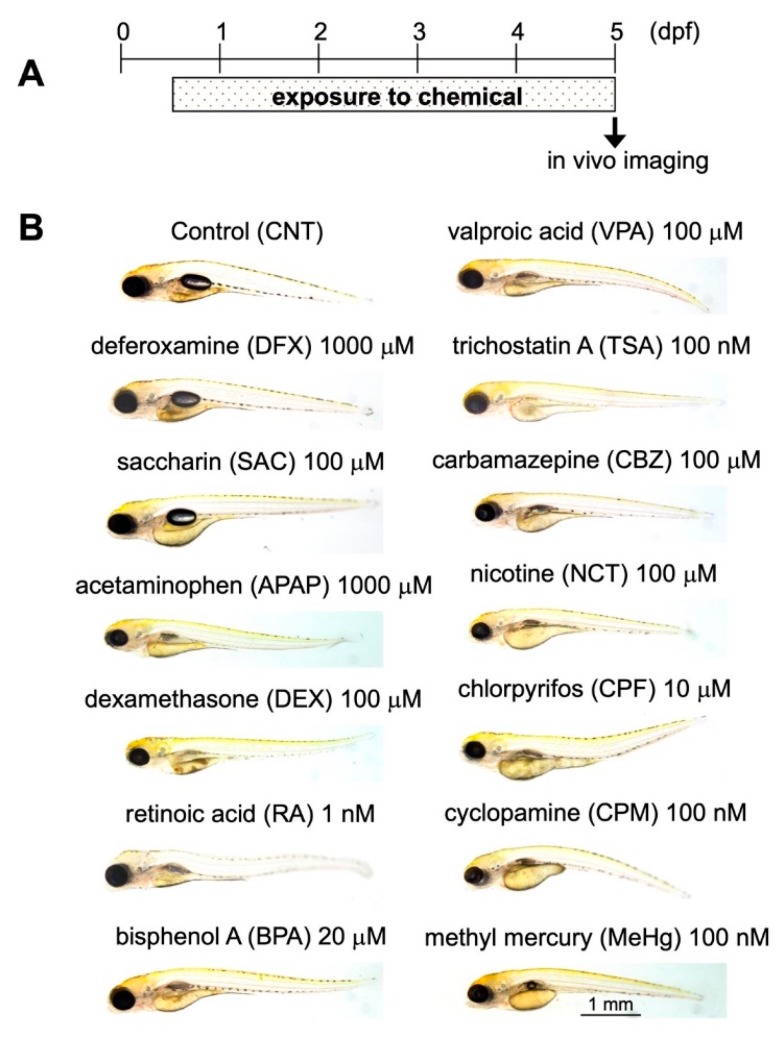
Bright-field images of triple-Tg zebrafish exposed to the maximum tolerable concentrations of chemicals during early development. (**A**) Experimental protocol. (**B**) Triple-Tg zebrafish were treated with the indicated chemicals at their maximum tolerable concentrations from 12 hpf to 5 dpf. The animals were then anesthetized and subjected to in vivo bright-field imaging using a stereomicroscope.

**Figure 4 pharmaceuticals-12-00145-f004:**
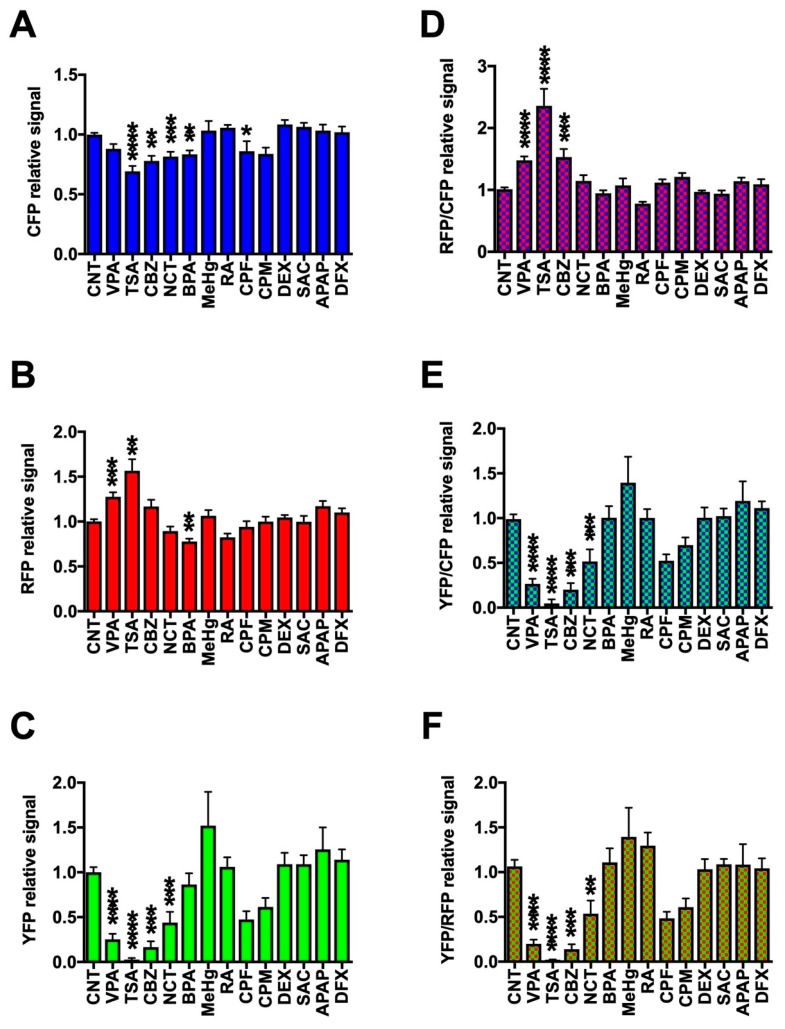
Quantification of in vivo fluorescence imaging of triple-Tg zebrafish exposed to chemicals at the maximum tolerable concentrations during early development. Triple-Tg zebrafish were treated as described for Figure 3 and subjected to in vivo fluorescence imaging at 5 dpf. The fluorescence signals for CFP (**A**), RFP (**B**), YFP (**C**), RFP/CFP ratio (**D**), YFP/CFP ratio (**E**), and YFP/RFP ratio (**F**) were quantified and normalized to the mean signals in the untreated control zebrafish group (CNT). * *p* < 0.05, ** *p* < 0.01, *** *p* < 0.001, **** *p* < 0.0001. Data are presented as the mean ± SEM of 4–77 zebrafish/chemical.

**Figure 5 pharmaceuticals-12-00145-f005:**
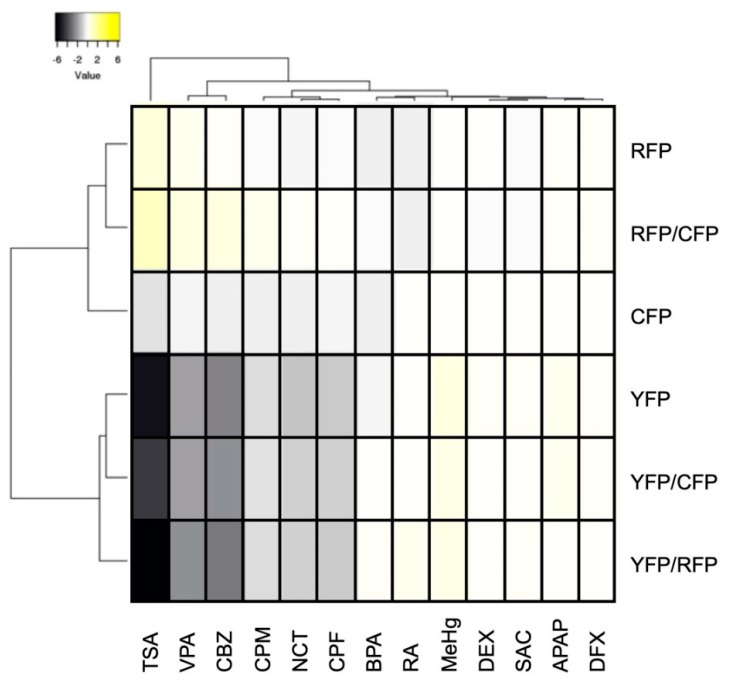
Hierarchical clustering of chemicals based on their effects on Cerulean (CFP), mCherry (RFP), and mCitrine (YFP) expression in triple-Tg zebrafish. The normalized score of six fluorescence parameters (CFP, RFP, YFP, RFP/CFP, YFP/CFP, and YFP/RFP) from triple-Tg zebrafish exposed to chemicals at their MTC from 12 hpf to 5 dpf were subjected to hierarchical clustering using Manhattan distance with average linkage.

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
