# Peer review of "Generation of a Triple-Transgenic Zebrafish Line for Assessment of Developmental Neurotoxicity during Neuronal Differentiation"

_pharmaceuticals, 2019, doi:10.3390/ph12040145_

Round 1

Reviewer 1 Report

This is an interesting work towards the development of a new model to analyze neurotoxicity during neuronal differentiation. The experimental design is appropriate. In addition, I appreciate that the authors described the limitations of their studying the discussion and I hope this triggers further studies in this area.

Before fully acceptance of the manuscript, the authors have to address the following questions:

Can you describe more in deep the chemical treatment? You added the chemicals to the medium, but did you remove the medium and add a fresh medium without chemicals before the analysis?  Statistical analysis, did you choose ANOVA because your data addressed the conditions to carry out a parametric test? Otherwise, you should use a non-parametric test.

Thank you very much.

Author Response

Reviewer 1

This is an interesting work towards the development of a new model to analyze neurotoxicity during neuronal differentiation. The experimental design is appropriate. In addition, I appreciate that the authors described the limitations of their studying the discussion and I hope this triggers further studies in this area. Before fully acceptance of the manuscript, the authors have to address the following questions:

Comments 1

Can you describe more in deep the chemical treatment? You added the chemicals to the medium, but did you remove the medium and add a fresh medium without chemicals before the analysis?

We have revised the manuscript to clarify that “…the resulting embryos (40 per well in 6-well plates) were exposed to serial dilutions of the chemicals from 12 hpf to 5 dpf without changing the medium.” (page 11, lines 449-451) and that “…the larvae were transferred to fresh 0.3× Danieau’s solution containing 2-phenoxyethanol (500 ppm) to be anesthetized…” (page 12, lines 459-460).

Comments 2

Statistical analysis, did you choose ANOVA because your data addressed the conditions to carry out a parametric test? Otherwise, you should use a non-parametric test.

Thank you for the suggestion. We have analyzed the data using Kruskal-Wallis test (a non-parametric test) and revised the manuscript (page 12, lines 476-479, Supplemental Table 1).

Reviewer 2 Report

The authors demonstrated DNT screening models using zebrafish. They examined 10 DNT positive compounds and 3 DNT negative compounds and tried to show the usefulness of the novel assay system. The data are potentially interesting, but there are some concerns.

1.The authors demonstrate that they could detect 8 of the 10 DNT compounds using zebrafish. According to the Fig. 4, when you used VFP signal, 6 compounds inhibited the neural differentiation. When you used rations, such as RFP/CFP, VPA, TSA, CBZ, NCT, CPF were detected, suggesting that 5 of the 10 DNT compounds. Therefore, the detection power were overdiscussed. The authors should revise the detection power in the manuscript. The authors tried to categorize the 13 compounds (Fig. 5).

2.How did the authors categorize the compounds? For example, APAP affected the RFP, YFP/CFP and YFP. However, the APAP and DFX were in a same category. It seems to me that SAC and DFX are in a same category. In addition, the category of MeHg seems to be strange. I did not understand how to perform hierarchical clustering. The authors should revised the manuscript and Fig.5.

3.According to the results, MeHg inhibited neural progenitor cells and the RGF and YFP signal are considered to be enhanced. However, only YFP was upregulated. The authors should explain the data more clearly. In the case of BPA, all signals (CPF, RFP, YFP) were inhibited. The authors should explain the data more clearly. The authors should discuss false negative compounds, such as RA, DEX, CPF. 

4.It is not clearly discussed why the zebrafish assay did not detect these false negative compounds. The authors should discusses these points more in  Otherwise, the readers would not understand the usefulness and limitations of the zebrafish assay. There is in vivo DNT guidelines by OECD and EPA. The authors should compare the in vivo data with the zebrafish data in the discussion section.

5.The authors selected 10 DNT compounds. Are there any reasons why the authors selected these positive compounds? It would be better to describe how to select the DNT compounds.

Author Response

Reviewer 2

The authors demonstrated DNT screening models using zebrafish. They examined 10 DNT positive compounds and 3 DNT negative compounds and tried to show the usefulness of the novel assay system. The data are potentially interesting, but there are some concerns.

Comments 1

The authors demonstrate that they could detect 8 of the 10 DNT compounds using zebrafish. According to the Fig. 4, when you used VFP signal, 6 compounds inhibited the neural differentiation. When you used rations, such as RFP/CFP, VPA, TSA, CBZ, NCT, CPF were detected, suggesting that 5 of the 10 DNT compounds. Therefore, the detection power were overdiscussed. The authors should revise the detection power in the manuscript.

In the revised manuscript, we have analyzed the data using Kruskal-Wallis test. We were able to detect significant effects of VPA, TSA, CBZ, NCT, BPA, and CPF on neuronal differentiation (Supplemental Table 1). We have revised the detection power (page 1, line 30, page 5, lines 200-218, page 9, lines 337-339).

Comments 2

How did the authors categorize the compounds? For example, APAP affected the RFP, YFP/CFP and YFP. However, the APAP and DFX were in a same category. It seems to me that SAC and DFX are in a same category. In addition, the category of MeHg seems to be strange. I did not understand how to perform hierarchical clustering. The authors should revised the manuscript and Fig.5.

We have revised Figure 5 and categorized the 13 chemicals (page 1, lines 31-35, page 5, lines 218-223, pages 9-10, lines 340-381).

Comments 3

According to the results, MeHg inhibited neural progenitor cells and the RGF and YFP signal are considered to be enhanced. However, only YFP was upregulated. The authors should explain the data more clearly. In the case of BPA, all signals (CPF, RFP, YFP) were inhibited. The authors should explain the data more clearly. The authors should discuss false negative compounds, such as RA, DEX, CPF.

We have revised the manuscript to clarify the effect of MeHg (page 5, lines 216-218, 222-223, page 10, lines 372-373), BPA (page 5, lines 202-206, 219-222 page 10, lines 364-367), RA (page 5, lines 216-218, page 10, lines 368-371), DEX (page 5, lines 216-218, page 10, lines 372-373), CPF (page 5, lines 202-204, page 9, lines 354-361), and CPM (page 5, lines 216-218, page 10, lines 361-363).

Comments 4

It is not clearly discussed why the zebrafish assay did not detect these false negative compounds. The authors should discusses these points more in. Otherwise, the readers would not understand the usefulness and limitations of the zebrafish assay. There is in vivo DNT guidelines by OECD and EPA. The authors should compare the in vivo data with the zebrafish data in the discussion section.

We have revised the manuscript to discuss the false-negative compounds compared with the studies in rodents (page 10, lines 372-378).

Comments 5

The authors selected 10 DNT compounds. Are there any reasons why the authors selected these positive compounds? It would be better to describe how to select the DNT compounds.

We have revised the manuscript to clarify that “We selected the 10 developmental neurotoxicants because they have been reported to affect neuronal differentiation” (page 5, lines 195-197).

Reviewer 3 Report

I have very few comments to this fine article.

As I am presenting for the authors

No doubt, this type of research has considerable potential for future research and applications.

Thus, my recommendation is acceptance after minor revisions

Please give an explanation of NOEC ( page 5 ) Non-observable ??

Give a textual explanation of Supplemental Table 1

Page 10, in Ethics Statement say " Nonetheless " in stead of " However "

Line 472 please describe how "selected " was done.

Questions

Could this model be used for environmental probes, in situ, on location ??

Microgravity experiments ?

Drug testing ?

Brain development ?

Author Response

Reviewer 3

I have very few comments to this fine article. As I am presenting for the authors No doubt, this type of research has considerable potential for future research and applications. Thus, my recommendation is acceptance after minor revisions

Comments 1

Please give an explanation of NOEC (page 5) Non-observable ??

We have revised the manuscript to clarify that “no observed effect concentration (NOEC) for lethality, which is the highest concentration of each chemical to which zebrafish are exposed that does not significantly affect the lethality compared with the controls” (page 5, lines 184-186).

Comments 2

Give a textual explanation of Supplemental Table 1.

We have revised the manuscript to give the textual explanation of Supplemental Table 1 (page 5, lines 200-218).

Comments 3

Page 10, in Ethics Statement say " Nonetheless " in stead of " However "

We have revised the manuscript as requested (page 10, line 404).

Comments 4

Line 472 please describe how "selected " was done.

We have revised the manuscript to clarify that “…were selected by in vivo fluorescence imaging at 5 dpf…” (page 11, lines 445-446).

Comments 5

Could this model be used for environmental probes, in situ, on location ?? Microgravity experiments ? Drug testing ? Brain development ?

We have revised the manuscript to clarify that “…using the triple-Tg zebrafish for drug discovery targeting neural regeneration and other studies focusing on neuronal differentiation” (page 10, lines 376-378).

Round 2

Reviewer 2 Report

Thank you for the revision of your manuscript. 

The manuscript has been much improved and seems to be suitable for publication.